# Development of S Haplotype-Specific Markers to Identify Genotypes of Self-Incompatibility in Radish (*Raphanus sativus* L.)

**DOI:** 10.3390/plants13050725

**Published:** 2024-03-04

**Authors:** Seong-Ho Heo, Su-Yeon Kim, Suk-Yeon Mo, Han-Yong Park

**Affiliations:** 1Department of Bioresources Engineering, Sejong University, Seoul 05006, Republic of Korea; heosh2409@sju.ac.kr (S.-H.H.); tndus7647@naver.com (S.-Y.K.); ysmo0708@naver.com (S.-Y.M.); 2Institute of Breeding Research, DASAN Co., Ltd., Pyeongtaek 17864, Republic of Korea

**Keywords:** radish (*Rahpanus sativus* L.), self-incompatibility, S haplotype, MSA

## Abstract

Radish (*Raphanus sativus* L.), a root vegetable belonging to the Brassicaceae family, is considered one of the representative crops displaying sporophytic self-incompatibility (SSI). The utilization of a self-incompatibility system in F_1_ breeding can improve the efficiency of cross-combinations, leading to a reduction in breeding time and aiding in the development of novel F_1_ varieties. The successful implementation of this system necessitates the rapid and accurate identification of S haplotypes in parental lines. In this study, we identified a total of nine S haplotypes among 22 elite radish lines through Sanger sequencing. Subsequently, we obtained sequences for showing a 95% similarity to nine S haplotypes, along with sequences identified by other researchers using BLAST. Following this, multiple sequence alignment (MSA) was conducted to identify *SRK* and *SLG* sequence similarities, as well as polymorphisms within the class I and II groups. Subsequently, S haplotype-specific marker sets were developed, targeting polymorphic regions of *SRK* and *SLG* alleles. These markers successfully amplified each of the nine S haplotypes. These markers will play a crucial role in the rapid and precise identification of parental S haplotypes in the radish F_1_ breeding process, proving instrumental in the radish F_1_ purity test.

## 1. Introduction

Radish (*Raphanus sativus* L.) is a root vegetable that belongs to the Brassicaceae family and is one of the most globally cultivated crops. The radish root is low in calories and rich in various minerals, particularly secondary metabolites such as polyphenols and glucosinolates, known for their anti-cancer effects [1,2]. Furthermore, radish is well known for exhibiting self-incompatibility and heterosis. Self-incompatibility facilitates outcrossing, enabling the maximal utilization of heterosis observed during radish F_1_ hybrid breeding.

Self-incompatibility (SI) is a mechanism that prevents self-fertilization and promotes outcrossing to maintain genetic diversity in plants [3]. Self-incompatibility can be classified into sporophytic self-incompatibility (SSI) and gametophytic self-incompatibility (GSI), depending on the timing of S gene expression [4,5,6]. Brassicaceae plants, including radish, exhibit sporophytic self-incompatibility (SSI) determined by the genotype of the pollen parent [7]. SSI is controlled by a single multi-allelic locus, called the S locus after ‘S’ for sterility [6,7,8].

Sporophytic self-incompatibility occurs in the stigma, involving the recognition and discrimination between self-pollen and non-self-pollen. This process selectively inhibits pollen tube development, thereby preventing self-fertilization [6,7,8]. There are three major genes that comprise the S locus: *SRK* (S locus receptor kinase), *SLG* (S locus glycoprotein), and *SCR/SP11* (S locus cysteine-rich protein/S locus protein 11) [9,10]. S locus receptor kinase (*SRK*) is specifically expressed in the stigma and acts as a maternal determinant [11]. *SRK* comprises three domains: the receptor domain (S domain), transmembrane domain, and serine/threonine kinase domain. The S domain is crucial in the process of self-pollen recognition, and its structural characteristics include 12 cysteine residues, potential N-glycosylation sites, and three hypervariable regions. The S domain of *SRK* and *SLG* alleles has the same structure features, and the sequence similarity of two genes is over 80% within S alleles [12,13]. S locus glycoprotein (*SLG*) is also specifically expressed in the stigma, and the function of the *SLG* gene has been reported to be crucial for stabilizing the self-incompatibility response. However, the exact function or mechanism remains unknown [11,14]. S locus cysteine-rich protein/S locus protein11 (*SCR/SP11*), specifically expressed in pollen, acts as a ligand in the allele-specific recognition between *SRK* and *SCR/SP11*. *SCR/SP11* activates a signal cascade mediating the SI response [8].

*SRK*, *SLG*, and *SCR/SP11*, involved in the S locus, are tightly linked to each other and tend to be inherited together in the next generation, forming what is called an S haplotype [15]. S haplotypes can be classified into two groups, class I and class II, based on amino acid sequence similarities of *SLG* and *SRK* [16]. The amino acid sequences of the S domain of *SRK* or *SLG* alleles show a 72% similarity within the same group and a similarity of 70% or less between different groups [17]. Additionally, *SLG* and the S domain of *SRK* alleles within the same S haplotype generally show high sequence similarity, especially in the class II group [18,19,20,21]. Between the two groups, there exists a complex genetic dominance relationship, where class I S haplotypes are generally dominant over class II S haplotypes in the pollen. On the other hand, in the stigma, codominance frequently occurs [20,22].

The SI system has been widely applied to F_1_ hybrid breeding, encompassing radish, cabbage, and other Brassica species. The utilization of the SI system has increased the efficiency of cross-combinations, shortening breeding time and facilitating the development of novel F_1_ varieties [23]. However, if parents share the same S haplotype and exhibit cross-incompatibility, it is difficult to produce a large scale of F_1_ hybrid seeds [24]. Therefore, it is necessary to select cross-combinations by early excluding radish breeding lines that exhibit the same S haplotype through the rapid and accurate identification of radish S haplotypes in the F_1_ hybrid breeding process.

Early studies on the identification of radish S haplotypes involved compatibility index (CI) analysis through test crosses, pollen tube examination, fluorescence analysis, and isoelectric focusing (IEF) gel examination [10,25,26]. However, these methods were labor-intensive, complex, and low-accuracy. With the subsequent advancement of PCR technology, Polymerase Chain Reaction–Restriction Fragment Length Polymorphism (PCR-RFLP) markers and Sequence-Characterized Amplified Region (SCAR) markers that specifically amplify radish *SRK* or *SLG* alleles have been developed and used for the identification of radish S haplotypes [27,28,29]. However, when using PCR-RFLP, the presence of identical restriction enzyme sites among some S haplotypes complicates the accurate identification process. With SCAR markers, the high sequence similarity between certain class I and II S haplotypes often leads to the co-amplification of several different S haplotypes [23,28,29].

Recently, the amplification of S locus genes using a universal primer set for class I and II S haplotypes, followed by Sanger sequencing to determine the nucleotide sequence, has become a common method. Subsequently, S haplotype determination involves comparing the obtained sequence with those already deposited in NCBI databases for similarity [23,24]. Even so, the use of gene nucleotide sequences and BLAST search requires the analysis of the nucleotide sequences of all parental lines, making it highly inefficient and unsuitable for large-scale S haplotype identification [23]. Despite the development of various methods, including molecular markers, there is still the challenge of finding an accurate and rapid method for identification of radish S haplotypes.

Furthermore, each researcher has independently assigned a nomenclature for radish S haplotypes in different countries, making the identification of radish S haplotypes more difficult [30,31]. In Japan, Niikura and Matsuura (1998) and Sakamoto (1998) identified nine *SLG* genes, naming S haplotypes as S201 and S1 to S8 [32,33]. Subsequently, Okamoto (2004) identified 10 SCR genes and 9 *SRK* genes, adopting the same nomenclature as Sakamoto (1998) [34]. In Korea, Lim (2002, 2004) identified 17 *SLG* gene sequences, naming them S1 to S26, while Park (2005) identified 5 SCR genes, naming them S1 to S5 [27,31]. Later, Kim (2016, 2019) named S haplotypes RsS-1 to RsS-31, considering combinations of *SRK*, SLL2, and SP6 genes [17]. More recently, Ni (2022) eliminated overlapping haplotypes based on BLAST analysis and newly named a total of 52 S haplotypes as NAU-S1 to NAU-S52 [23]. Given the various nomenclatures in current databases, it is essential to establish specific criteria for global S haplotype classification to facilitate the sharing of information on individual S haplotype characteristics [31].

Therefore, in this study, we aimed to overcome the limitations of existing S haplotype markers and develop an accurate and rapid method for radish S haplotype identification. Firstly, we classified and compared S haplotypes named differently by Korean, Chinese, and Japanese researchers in the NCBI database according to specific criteria. Secondly, using 22 radish breeding lines, mainly utilized for radish F_1_ hybrid breeding in Korea and Japan, among other regions, and stabilized through more than seven generations of self-pollination, we developed radish S haplotype-specific markers based on the nucleotide sequences of *SRK* and *SLG* alleles. The S haplotype-specific markers developed in this study are expected to be effectively used for identifying the S haplotypes of parental lines, selecting cross-combinations for F_1_ hybrid breeding, and performing seed purity tests during the F_1_ hybrid breeding process.

## 2. Results

### 2.1. Amplification of SRK, SLG Alleles Using Universal Primers

A total of three class I *SRK* and *SLG* universal primer sets, along with four class II *SRK* and *SLG* universal primer sets, were used to amplify the kinase domain of *SRK* and *SLG* alleles of 22 radish breeding lines (‘SJ-1~22’).

In 14 radish breeding lines (‘SJ-1~14’), the class I kinase domain of *SRK* was amplified using the UV*SRK*-F + UV*SRK*-R primer combination. Additionally, class I *SLG* alleles were amplified with the *SLG*-I-F + *SLG*-I-R and PS22 + *SLG*-I-R primer combinations, resulting in a total band size of 1000 to 1500 bp (Appendix A).

In eight radish breeding lines (‘SJ-15~22’), the class II kinase domain of *SRK* was amplified using the KS2 + KA2 primer combination. Additionally, class II *SLG* alleles were amplified with the *SLG*-II-F + *SLG*-II-R, Rs9 *SLG*-F + Rs9 *SLG*-R, and UV*SLG*II-F + UV*SLG*II-R primer combinations, resulting in a total band size of 800 to 1500 bp (Appendix A).

### 2.2. Identification of S Haplotypes Based on BLAST Search

We performed BLAST search using *SRK* and *SLG* alleles amplified by universal primer sets from ‘SJ-1~22’ to identify the corresponding S haplotypes. These sequences were compared to previously published S haplotypes in the NCBI database or from other researchers. As a result, it was determined that the 22 radish breeding lines possessed a total of nine S haplotypes (Table 1). Nucleotide sequence information is provided in the Appendix A, and if two or more radish lines had the same S haplotype, only the sequence obtained from one of them was indicated.

A total of 14 radish breeding lines (‘SJ-1~14’) were classified as the class I S haplotype. ‘SJ-1~4’ were identified as S1 (Lim), since the kinase domain of *SRK* and *SLG* alleles showed 100% and 99% identities to *SRK*1(Lim) and *SLG*1(Lim), respectively. ‘SJ-5~6’ were identified as S8 (Lim) due to a 100% identity in the kinase domain of the *SRK* allele with *SRK*8 (Lim), *SRK*19 (Kim), and *SRK*-19 (Haseyama), along with a 100% identity in the *SLG* allele with *SLG*-19 (Haseyama). S8 (Lim) has been reported as the same S haplotype as S19 (Kim) and RsS-19 (Haseyama) [31]. Additionally, the kinase domain of the ‘SJ-5~6’ *SRK* allele showed a 97% identity with *SRK*-18 (Haseyama). ‘SJ-7~9’ were identified as S10 (Lim), since the kinase domain of *SRK* and *SLG* alleles showed a 99.9% identity to *SRK*10 (Lim) and *SLG*10 (Lim), respectively. There were 11 other *SRK* sequences showing 98 to 99% identities to the kinase domain of the ‘SJ-7~9’ *SRK* allele (Table 1, Appendix A). ‘SJ-10~11’ were identified as S16 (Lim) due to a 99.9% identity in the kinase domain of the *SRK* allele with *SRK*16 (Lim) and *SRK*-22 (Haseyama), along with a 100% identity in the *SLG* allele with *SLG*-22 (Haseyama). S16 (Lim) has been reported as the same S haplotype as RsS-22 (Haseyama) [31]. ‘SJ-12~14’ were identified as S18 (Lim) due to a 99.9% identity in the kinase domain of the *SRK* allele with *SRK*18 (Lim) and *SRK*-6 (Haseyama), along with a 100% identity in the *SLG* allele with *SLG*18 (Lim) and *SLG*-6 (Sakamoto). S18 (Lim) has been reported as the same S haplotype as RsS-6 (Sakamoto) [31].

A total of eight radish breeding lines (‘SJ-15~22’) were classified as the class II S haplotype. ‘SJ-15~17’ were identified as S4 (Lim) due to 99% and 100% identities in the *SLG* allele with *SLG*4 (Lim) and *SLG*-26 (Lim). S4 (Lim) has been reported as the same S haplotype as RsS-26 (Haseyama) [31]. Additionally, the ‘SJ-15~17’ *SLG* allele showed a 97% identity with *SLG*-11 (Haseyama). ‘SJ-18~20’ were identified as S5 (Lim) due to a 99% identity in the kinase domain and S domain of the *SRK* allele with *SRK*-5 (Wang). S5 (Lim) has been reported as the same S haplotype as S-5 (Wang) [24]. Also, the kinase domain of the ‘SJ-18~20’ *SRK* allele showed a 99% identity with *SRK*-9 (Wang) and *SRK*6 (*R. raphanistrum*). Additionally, the S domain of the ‘SJ-18~20’ *SRK* allele showed 97 to 99% identities with 10 other class II S domains of *SRK* and *SLG* alleles (Table 1). ‘SJ-21’ was identified as S21 (Lim) due to a 99% identity in the kinase domain of the *SRK* allele with *SRK*-9 (Wang) and a 100% identity in the *SLG* allele with *SLG*21 (Lim), *SLG*-9 (Haseyama), and *SLG*-9 (Wang). S21 (Lim) has been reported as the same S haplotype as RsS-9 (Haseyama) and S-9 (Wang) [24]. Also, the kinase domain of the ‘SJ-21’ *SRK* allele showed a 99% identity with *SRK*-5 (Wang) and *SRK*6 (*R. raphanistrum*). Additionally, the ‘SJ-21’ *SLG* allele showed 94 to 100% identities with nine other class II S domains of *SRK* and *SLG* alleles (Table 1). ‘SJ-22’ was identified as S26 (Lim) due to a 100% identity in the *SLG* allele with *SLG*26 (Lim) and *SLG*-29 (Haseyama). S26 (Lim) has been reported as the same S haplotype as RsS-29 (Haseyama) [31].

### 2.3. Multiple Sequence Alignment (MSA) of SRK, SLG Alleles

Multiple sequence alignment (MSA) of nucleotide and amino acid sequences was performed to develop S haplotype-specific markers targeting polymorphic regions of *SRK* and *SLG* alleles. In this alignment, we used all the S haplotype sequences registered by Korean, Chinese, and Japanese researchers. This included 7 kinase domains of *SRK* and 10 *SLG* alleles from Lim’s group [27]; 10 kinase domains of *SRK* alleles from Kim’s group [17], 3 kinase domains and 3 S domains of *SRK* alleles from Wang’s group [24]; 7 *SLG* alleles from Sakamoto’s group [33]; along with 8 kinase domains, 3 S domains of *SRK* alleles, and 4 *SLG* alleles from Haseyama’s group [31]. Only one sequence of the S haplotype was used when the same S haplotype was designated differently.

A total of 19 class I kinase domains of *SRK* sequences were used for the alignment of the class I kinase domain of *SRK* alleles. The similarity between the class I kinase domain of *SRK* nucleotide sequences ranged from 83.86 to 91.85%. All exon regions (4th to 7th) were conserved relative to intron regions (4th to 7th). In the 6th and 7th exons, some nucleotide polymorphisms, including insertions/deletions (InDels), were detected. In the sixth exon region, a small gap (13 bp) was observed. Additionally, a large gap (41 to 49 bp) and a small gap (19 to 29 bp) were observed in the 4th and 6th intron regions, excluding *SRK*-KD18 (Haseyama) and *SRK*-KD1 (Haseyama) (Appendix A).

A total of 15 class I *SLG* sequences were used for the alignment of class I *SLG* alleles. The similarity between class I *SLG* nucleotide sequences ranged from 84 to 88.94%. Many single nucleotide polymorphisms (SNPs) were observed in hypervariable regions I to III (HV-I to III). Especially, a small gap (6 bp) was observed in the downstream region of HV-II, excluding *SLG*4 (Sakamoto). The deduced amino acid sequences of 14 class I *SLG* alleles revealed characteristics as previously reported, including 12 conserved cysteine residues, the potential N-glycosylation sites, and three hypervariable regions (Figure 1 and Appendix A).

A total of nine class II kinase domains of *SRK* sequences were used for the alignment of the class II kinase domain of *SRK* alleles. The similarity between the class II kinase domain of *SRK* nucleotide sequences ranged from 90.17 to 99.17%. Notably, the *SRK*26-KD (Wang) nucleotide sequence showed a high similarity (99.17%) with *SRK*14-KD (Kim). As observed in the case of the class I kinase domain of *SRK* alleles, all exon regions were conserved relative to intron regions. On the other hand, all exon regions showed high similarity within the same group of the class II kinase domain of *SRK* alleles. Some single nucleotide polymorphisms (SNPs) and insertions/deletions (InDels) were observed in the sixth intron region; particularly, a small gap (14 bp) was only detected in *SRK*-5 (Wang) and *SRK*-9 (Wang) (Appendix A).

Due to the high similarity between the class II *SLG* and the S domain of *SRK* sequences, a total of 12 sequences including *SLG* and the S domain of *SRK* alleles were used for the alignment. The similarity between class II *SLG* nucleotide sequences was 90 to 97.05%, and when including the S domain of *SRK* sequences, they showed 87.5 to 99.16%. In the upstream region of HV-I, small gaps (9, 12 bp) were observed. In the deduced amino acid sequences of the class II *SLG* and the S domain of *SRK* alleles, 12 conserved cysteine residues were detected at the same positions as those in class I *SLG* alleles, excluding *SLG*4 and 21 (Lim) and *SRK*-5 (Wang). Additionally, the potential N-glycosylation sites and three hypervariable regions were found to be located at the same positions (Figure 2 and Appendix A).

**Figure 1 plants-13-00725-f001:**
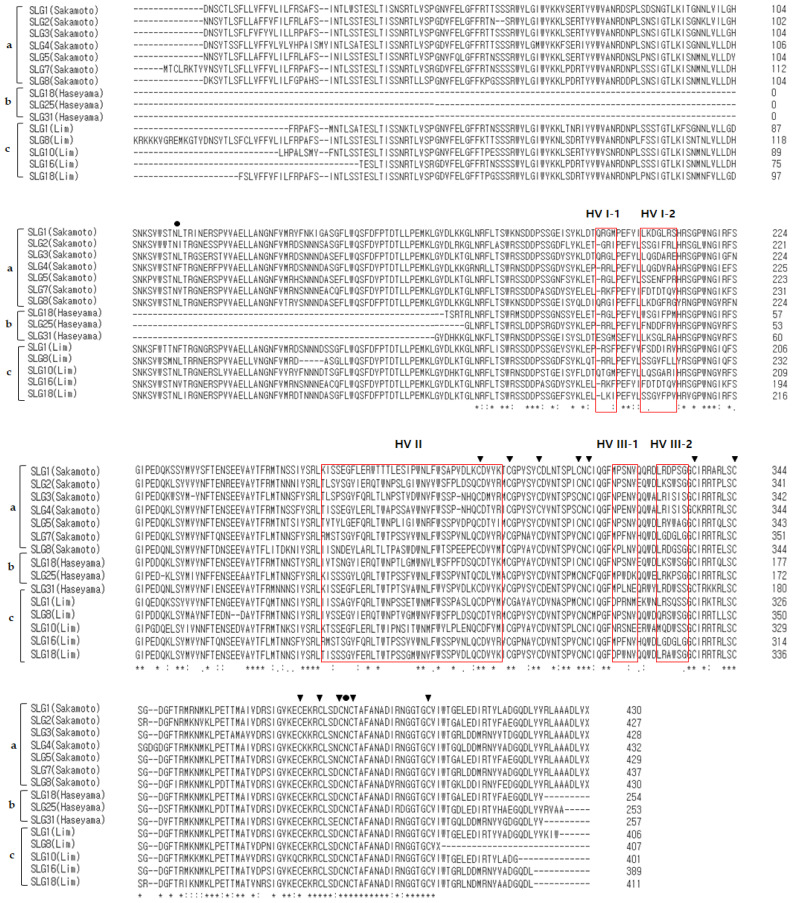
Multiple sequence alignment of the amino acid sequences of the 15 class I S haplotype *SLG* alleles using Clustal Omega program on EMBL-EBI site; a: S haplotype published by Sakamoto [33]; b: S haplotype published by Haseyama [31]; c: S haplotype published by Lim [27]; red box: hypervariable regions I to III of the *SLG*; filled circle: N-linked glycosylation site; filled triangle: conserved cysteine residues; asterisk: conserved amino acid residues.

**Figure 2 plants-13-00725-f002:**
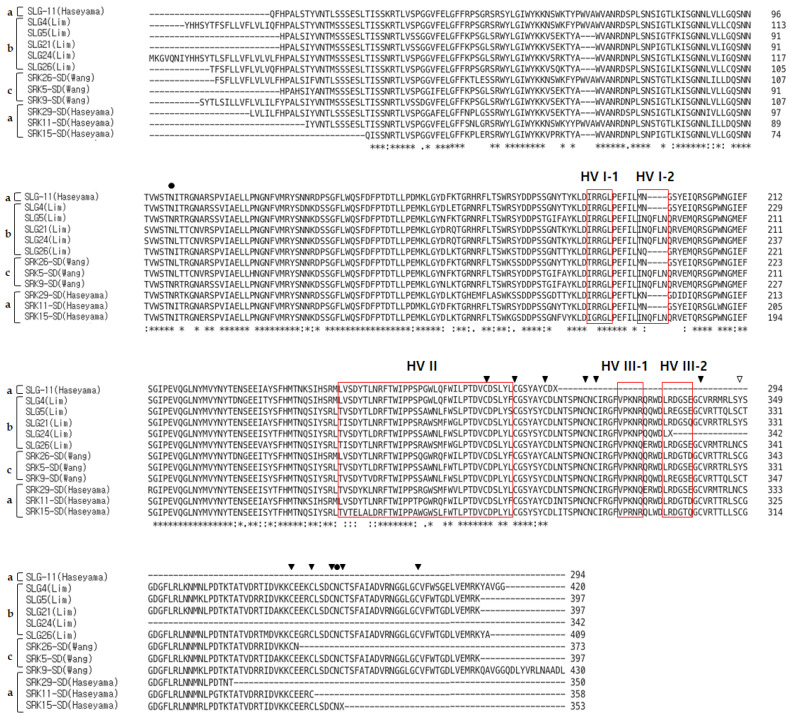
Multiple sequence alignment of the amino acid sequences of the 12 class II S haplotype *SLG* alleles and the S domain of *SRK* alleles using Clustal Omega program on EMBL-EBI site; a: S haplotype published by Haseyama [31]; b: S haplotype published by Lim [27]; c: S haplotype published by Wang [24]; red box: hypervariable regions I to III of the *SLG*; filled circle: N-linked glycosylation site; filled triangle: conserved cysteine residues; empty triangle: the site where non-synonymous mutation was inferred to have occurred; asterisk: conserved amino acid residues.

### 2.4. Development of S Haplotype-Specific Markers

Based on the results of class I and II *SRK* and *SLG* sequence alignment, we identified regions showing single nucleotide polymorphisms (SNPs) or small insertions/deletions (INDELs) within class I and II groups. Targeting these regions, we developed S haplotype-specific markers to selectively amplify the nine S haplotypes present in 22 radish breeding lines (‘SJ-1~22’) (Table 4 and Appendix A).

A total of four kinase domains of *SRK* primer sets were designed, with each primer set amplifying specific regions of *SRK*1 (Lim), *SRK*8 (Lim), *SRK*10 (Lim), and *SRK*18 (Lim) kinase domains, ranging from the 4th to 7th exon of *SRK* alleles. The expected PCR product size using these primer sets was 105 to 764 bp, designed to exhibit length polymorphism with each primer set (Table 2)

Additionally, nine *SLG* primer sets were designed, each amplifying specific regions of *SLG*1 (Lim), *SLG*4 (Lim), *SLG*5 (Lim), *SLG*8 (Lim), *SLG*10 (Lim), *SLG*16 (Lim), *SLG*18 (Lim), *SLG*21 (Lim), and *SLG*26 (Lim), ranging from HV-I to III of *SLG* alleles. The expected PCR product size using these primer sets was 131 to 529 bp, similar to the kinase domain of *SRK* alleles, designed to show length polymorphism with each primer set (Table 2).

To verify the specificity of the kinase domain of *SRK* and *SLG* primer sets developed in this study, genotyping was conducted on 22 radish breeding lines (‘SJ-1~22’) using the kinase domain of *SRK* and *SLG* primer sets. Consequently, all 5 *SRK* primer sets and 9 *SLG* primer sets selectively amplified radish breeding lines possessing a distinct S haplotype, and the predicted length of the PCR product from in silico analysis aligned with the observed results (Figure 3 and Figure 4). Furthermore, it was noted that the drawbacks, such as the amplification of nonspecific S haplotypes, which arose from the previously developed SCAR marker, did not occur in the S haplotype-specific markers developed in this study.

## 3. Discussion and Conclusions

To develop S haplotype-specific markers, it was necessary to determine the sequences of S alleles (*SRK*, *SLG*) for a specific S haplotype in radish breeding lines (SJ-1~22). For this purpose, *SRK* and *SLG* genes were amplified using three types of class I universal primer sets and four types of class II universal primer sets. Due to the sequence variations in S alleles, amplifying all S alleles with a limited combination of primers posed a challenge. Therefore, we used various primer combinations developed in previous studies [27,33,35,36,37,38,39]. Although PCR amplification was performed using seven class I and II universal primers, it was difficult to identify S haplotypes due to inaccurate amplification of the non-target genes in some cases. For instance, when performing PCR amplification using three universal class II *SLG* primer sets in ‘SJ-18~20’, it was observed that PCR amplification occurred for the S domain of the *SRK* allele instead of the *SLG* allele. These results are consistent with previous reports indicating that the sequences of the S domain of *SRK* and *SLG* alleles within the class II group were highly similar [21,24,40]. Additionally, when using the universal class II kinase domain of *SRK* primer sets in ‘SJ-15~17’ and ‘SJ-22’, other regions with the same primer binding sites were co-amplified, leading to sequencing errors (mixed signal). As a result, the exact kinase domain of *SRK* sequences of ‘SJ-15~17’ and ‘SJ-22’ could not be identified.

As a result of classifying and comparing class I S haplotypes using BLAST analysis, within the class I S haplotype group, including *SRK* and *SLG* alleles of ‘SJ-1~14’, along with those obtained through BLAST search in the NCBI database, there were numerous cases where the kinase domain of *SRK* alleles exhibited significant similarity to each other. This result corresponds with the observations during the previous identification of class I S haplotypes in the Brassicaceae family, including radishes [24,29]. The kinase domain of *SRK* alleles has been reported to be the region where recombination could occur, and the observed high similarity within the class I kinase domain of *SRK* alleles might be attributed to genetic recombination events [41]. In contrast to previous reports suggesting recombination in *SLG* alleles, we did not observe high similarity within class I *SLG* alleles [42].

Furthermore, based on Lim’s nomenclature, we compared a total of five class I S haplotypes identified in 14 radish breeding lines (SJ-1~14) with those investigated by other researchers. Through this comparison, we could classify sequences that were similar or identical (Table 1). Consistent with previously reported research, observations revealed that S8 (Lim) corresponds to RsS-19 (Haseyama, Kim), S16 (Lim) corresponds to RsS-22 (Haseyama), and S18 (Lim) corresponds to RsS-6 (Sakamoto, Haseyama) [27,31,33,39]. When BLAST search was conducted using kinase domain sequences of *SRK*10 (Lim) and *SRK*16 (Lim) as queries, 11 kinase domains of *SRK* sequences showing high similarity (98 to 99%) were identified. Subsequent sequence alignment analysis of *SRK*10 and 16 (Lim) kinase domains with these similar kinase domains revealed nearly identical nucleotide sequences, with only a few single nucleotide polymorphisms (SNPs) (Appendix A). Although the *SLG*16 (Lim) nucleotide sequence identified in ‘SJ-10~11’ and *SLG*-7 (Sakamoto) had a 99.9% identity, S16 (Lim) has not been reported to be the same as RsS-7 (Sakamoto). Consequently, a comparison of the total length of two *SLG* alleles or S domains of *SRK* alleles was necessary to reclassify the two S haplotypes.

As a result of classifying and comparing class II S haplotypes using BLAST analysis, in the class II S haplotype group, including *SRK* and *SLG* alleles of ‘SJ-15~22’, along with those obtained through BLAST search in the NCBI database, the kinase domains of *SRK* alleles as well as *SLG* alleles were highly similar to each other, unlike the class I S haplotype group. Previous studies have reported that the kinase domain of *SRK* and *SLG* alleles is the region where recombination could occur [41,43]. The class II S haplotype group is inferred to be more recently diverged than the class I S haplotype group, as indicated by the high similarity within the class II S haplotype group [44]. According to the Brassica SI evolutionary model proposed by Uyenoyama, the class II S haplotype group might invade populations at lower rates than the class I S haplotype, leading to a decrease in the occurrence of mutations and divergence within the class II group [45].

Moreover, based on Lim’s nomenclature, we compared a total of four class II S haplotypes identified in eight radish breeding lines (SJ-15~22) with those investigated by other researchers. Through this comparison, we could classify sequences that were similar or identical (Table 1). Consistent with previously reported research, observations have revealed that S4 (Lim) corresponds to RsS-26 (Haseyama), S5 (Lim) corresponds to RsS-5 (Wang), S21 (Lim) corresponds to RsS-9 (Wang, Haseyama), and S26 corresponds to RsS-26 (Haseyama) [24,27,31]. The *SLG*4 (Lim) sequence identified in ‘SJ-12~14’ showed a 97% similarity with *SLG*-11 (Haseyama), but S4 (Lim) has not been reported to be the same as RsS-11 (Haseyama). The S domain of the *SRK*5 (Lim) sequence exhibited a high similarity with *SLG*21 (Lim) and *SLG*24 (Lim) at 95% and 96%, respectively. This is presumed to result from the gene duplication of the S domain of *SRK* alleles, intergenic recombination within *SLG* alleles, and gene conversion between the S domain of *SRK* and *SLG* alleles [18,42,46,47,48]. The kinase domain of *SRK*5 (Lim) and *SRK* 21 (Lim) sequences not only showed similarity with each other but also revealed high similarity with the kinase domain of the wild radish *SRK* allele (99%). It has been reported that class II S haplotypes of cultivated cabbage have been identified in wild species of the cabbage group [49]. This implies that nucleotide sequences more similar or identical to class II S haplotypes of cultivated radish may be discovered in wild radish. High-similarity regions with *SLG*5 (Lim) and *SLG*21 (Lim) were searched on radish chromosomes 7 and 8 (OY743213, LR778317). The S locus is a single gene locus located on chromosome 7 [50]. When performing local BLAST with the *SLG*21 (Lim) sequence as a query on the radish genome ‘QZ-16’ (GCA_902824885.1), including chromosomes 7 (LR778316) and 8 (LR778317), sequences corresponding to *SLG*21 (Lim) were only found on chromosome 8, with none matching on chromosome 7 (Appendix A). Therefore, it is highly probable that the two regions searched on chromosome 8 (LR778317) were the S domain of *SRK* and *SLG* alleles rather than S homologs. This issue was inferred to result from errors during the chromosome-scale assembly process, and it could potentially lead to confusion in identifying radish S haplotypes through BLAST search.

In this study, multiple sequence alignment was performed using all the S haplotype sequences identified until now by Korean, Chinese, and Japanese researchers. Through this, variations in *SRK* and *SLG* alleles for each class group were detected.

When multiple sequence alignment (MSA) was performed using *SRK*/*SLG* sequences of class I S haplotypes, they exhibited a similarity ranging from 83.86% to 91.85%. This closely approximates the similarity range of 80% to 90% reported in previous research [51]. A gap of 19 to 29 bp was observed only in the sixth intron of the *SRK*6 (Lim) kinase domain and an insertion of 6 bp was observed only in the downstream region of HV-III in *SLG*-4 (Sakamoto). It was thought that these regions could be used to design candidate S6 (Lim) and S4 (Sakamoto) haplotype-specific markers.

When multiple sequence alignment (MSA) was conducted on the *SRK*/*SLG* sequences of class II S haplotypes, they exhibited a similarity ranging from 90% to 99.17%. Additionally, a remarkably high similarity (87.5% to 99.16%) was observed between the *SLG* gene and the *SRK* S domain. This observation is consistent with previous research indicating that within the class II group, *SRK*/*SLG* sequences show higher similarity compared to the class I group [44,48]. Notably, *SLG*5 (Lim) and the S domain of *SRK*21 (Lim) showed the highest similarity (99.16%), although they were different S haplotypes. This result elucidated why the two S haplotypes (S5, S21) were co-amplified when previously developed SCAR markers were used.

Additionally, we observed tyrosine residue in HV-III of *SLG*4 and 21 (Lim) and the S domain of *SRK*21 (Lim). It was presumed that non-synonymous mutations might have occurred at that site, substituting the 12th cysteine residue with tyrosine. Based on the observation that mutations have occurred not only in *SLG* alleles but also in the S domain of *SRK* alleles, it is considered necessary to further study whether the SNP mutation at that location affects the main function of the S domain of *SRK* alleles, which perceives and rejects self-pollen in the stigma.

Based on the results of BLAST search and sequence alignment, there was a potential risk of nonspecific amplification of the kinase domain of *SRK* alleles. For a more accurate identification of the target S haplotype, it might be preferable to prioritize *SLG* allele-specific primer sets over the kinase domain of *SRK* primer sets.

When using S haplotype-specific markers developed in this study, it will be possible to overcome the shortcomings of previously developed SCAR markers, such as the amplification of nonspecific S haplotypes. These markers can be used for the more accurate and rapid amplification of target S haplotypes in radish (Figure 3 and Figure 4). These markers are anticipated to aid in identifying the S haplotypes of parental lines and selecting cross-combinations during radish F_1_ breeding processes. Furthermore, these markers are expected to be effectively applied in the F_1_ seed purity test.

To identify additional S haplotypes rapidly and accurately, in addition to the nine S haplotypes already identified, it is considered imperative to develop other S allele-specific markers based on the methods applied in this study.

## 4. Materials and Methods

### 4.1. Plant Materials

The 22 elite cultivated radish lines (‘SJ-1~22’), serving as parental lines for F_1_ hybrid breeding in Korea, Japan, and other regions, were used as plant materials in this study [52]. The plant materials were rendered homozygous through self-pollination via bud pollination for more than 7 generations at a plant breeding house in Gyeonggi-do, Yeoju, Yanggui-ri, Republic of Korea (Table 3).

### 4.2. Amplification and Sequencing of SRK, SLG Alleles

#### 4.2.1. Extraction of Genomic DNA and Amplification of *SRK*, *SLG* Alleles

Genomic DNA was extracted from seedling leaves (15 to 20 days after germination) of each radish breeding line using the CTAB method [53]. The concentration and purity of all extracted genomic DNAs were assessed using a nanodrop machine (DeNovix Co., Wilmington, DE, USA).

For the PCR amplification of *SRK* and *SLG* nucleotide sequences of ‘SJ-1~22’, 7 class I, II *SRK* and *SLG* universal primer sets were used in this study (Table 1 and Appendix A). The UV*SLG*II primer set was designed to amplify the greatest region of the *SLG*21 (Lim) nucleotide sequence with the multiple sequence alignment of 12 class II *SLG* alleles and S domains of *SRK* sequences identified from radish. This primer set targeted the 5’ and 3’ class II *SLG*-conserved nucleotide sequences. PCR amplification was performed in a 15 μL reaction mixture, consisting of 7.5 μL Dyne Ready 2X-GO (Star Plus Taq with Dye; Dynebio Co., Gyeonggido, Republic of Korea), 1.5 μL 5X Tune-up solution, 1.5 μL forward primer (5 μM), 1.5 μL reverse primer (5 μM), 1.5 μL template DNA (50 ng/μL), and 1.5 μL distilled water. The PCR amplification conditions involved an initial denaturation at 95 °C for 3 min and a final elongation at 72 °C for 5 min, with the optimal annealing temperature applied to each primer (Table 4).

#### 4.2.2. Identification of S Haplotypes Based on BLAST Search

The PCR products obtained by amplifying *SRK* and *SLG* alleles of 22 radish breeding lines were separated by electrophoresis on a 2% agarose gel stained with 7.2 μL of EcoDye (Biofact Co., Daejeon, Republic of Korea) in 360 mL of 0.5X TBE buffer at 250 V for 40 min and visualized using an Ultraviolet (UV) Transilluminator. Subsequently, gel extraction was carried out to obtain accurate nucleotide sequence information for *SRK* and *SLG* alleles. Slices of the agarose gel containing the amplified target *SRK* and *SLG* alleles were cut to approximately 0.2 to 0.3 g, transferred to a 2 mL micro-centrifuge tube, and purified using the LaboPass Gel and PCR Clean-up Kit (Cosmogenetech Co., Seoul, Republic of Korea). Following purification, Sanger sequencing was performed by Cosmogenetech Co. (Seoul, Republic of Korea). To address base noise, sequencing was performed twice, using both forward and reverse primers. Any base with a base quality value (QV) of 20 or less was excluded from the sequencing analysis.

To determine the S haplotypes of the 22 elite breeding lines (‘SJ-1~22’), the sequencing data were compared with other kinase domains and S domains of *SRK* and *SLG* alleles registered in the National Center for Biotechnology Information (NCBI) or other database using the Basic Local Alignment Search Tool (BLAST) program.

Since the nomenclature of S haplotype differed for each researcher in each country and was not unified, the S haplotypes were classified based on the nomenclature proposed by Lim [27].

### 4.3. Multiple Sequence Alignment (MSA)

To develop S haplotype-specific markers, we attempted to identify target regions of *SRK* and *SLG* alleles that showed sequence or structural variation. We performed multiple sequence alignment (MSA) for both nucleotide and amino acid sequences. This analysis involved the S haplotypes amplified by PCR from 22 radish breeding lines, with S haplotypes showing more than 94% similarity through a BLAST search, and additional S haplotypes identified by researchers in Korea [17,27], China [24], and Japan [31,33].

Using BLAST, we acquired *SRK* and *SLG* nucleotide sequences with more than 94% similarity to *SRK* and *SLG* alleles of ‘SJ-1~22’. Subsequently, MSA of both nucleotide and amino acid sequences was performed using EMBL-EBI (European Bioinformatics Institute), MAFFT version 7.0, and MEGA version 11.0 programs. Hypervariable regions 1 to 3 (HV-I to III) of *SRK* and *SLG* alleles were identified through the EMBL-EBI sixpack tool.

### 4.4. Development of S Haplotype-Specific Markers

Based on the results of MSA, S haplotype-specific markers were developed, targeting specific regions that show polymorphism in *SRK* and *SLG* alleles of ‘SJ-1~22’. The Primer-BLAST program was used, with the primer length set between 20 and 30 bp, the melting temperature (Tm) value ranging from 52 to 66 °C, and the GC content around 60%. The annealing temperature was calculated using NEB Tm calculator version 1.15.0.

First, candidate S haplotype-specific markers that selectively amplified the kinase domain of *SRK* and *SLG* of each ‘SJ-1~22’ in silico were chosen. Primer sets that exhibited overlapping binding sites, low polymorphism, or amplified regions of 100 bp or less were discarded. Subsequently, PCR amplification and gel electrophoresis were performed on the 22 elite breeding lines (‘SJ-1~22’) using the selected S haplotype-specific markers to assess whether they specifically amplified the corresponding S haplotypes.

## Figures and Tables

**Figure 3 plants-13-00725-f003:**
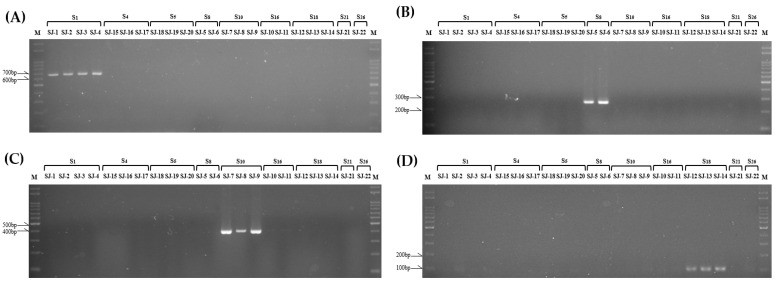
PCR result of 22 radish breeding lines (‘SJ-1~22’) using developed kinase domain of *SRK* (*SRK*-KD)-specific primer sets in this study. ‘S1~S26’ are the S haplotypes that each line has. (**A**): *SRK*1 (665 bp); (**B**): *SRK*8 (271 bp); (**C**): *SRK*10 (411 bp); (**D**): *SRK*18 (105 bp).

**Figure 4 plants-13-00725-f004:**
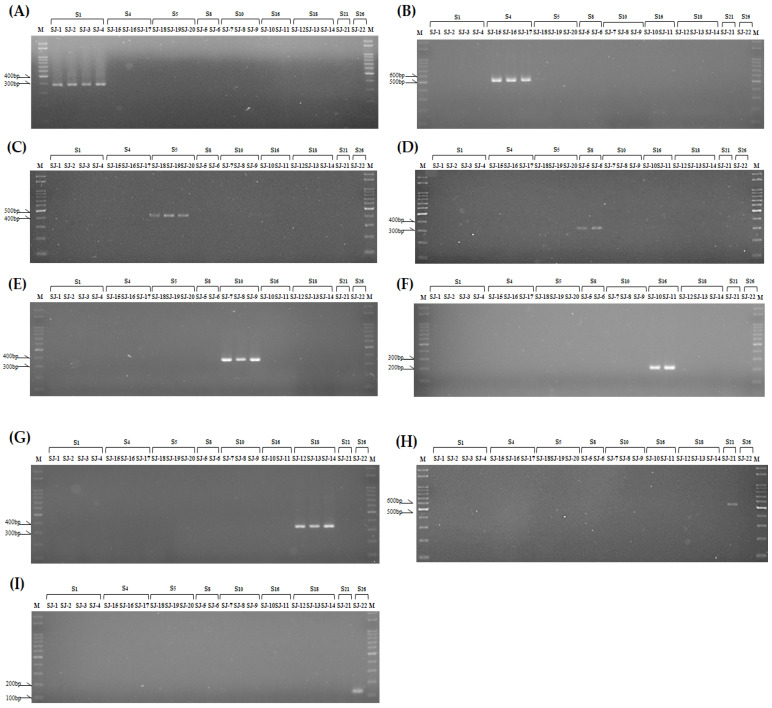
PCR result of class I radish breeding lines (‘SJ-1~22’) using developed *SLG*-specific primer sets in this study. Each number is the S haplotype that each S homozygous line has. (**A**): *SLG*1 (388 bp); (**B**): *SLG*4 (529 bp); (**C**): *SLG*5 (204 bp); (**D**): *SLG*8 (314 bp); (**E**): *SLG*10 (369 bp); (**F**): *SLG*16 (216 bp); (**G**): *SLG*18 (353 bp); (**H**): *SLG*21 (445 bp): (**I**): *SLG*26 (131 bp).

**Table 1 plants-13-00725-t001:** Identification and comparison of S haplotypes using BLAST program; Acc.: accession name of radish breeding lines (‘SJ-1~22’) used in this study; NCBI Acc.: accession number of genes registered in NCBI database; bold letters: S haplotype with best hit in result of BLAST search.

Acc.	S Haplotype	S Alleles	Tool	Gene	NCBI Acc.	Iden (%)	Query Cov
SJ-1~4	S1 (Lim)	*SRK*-KD (1063 bp)	BLASTn	***SRK*1 (Lim)**	**AY052579**	**100%**	**99%**
		*SLG* (1220 bp)	BLASTn	***SLG*1 (Lim)**	**AY052572**	**99%**	**99%**
SJ-5~6	S8 (Lim)	*SRK*-KD (1133 bp)	BLASTn	***SRK*8 (Lim)**	**AY052583**	**100%**	**100%**
				***SRK*19**	**KX961713**	**100%**	**100%**
				*SRK*-19	LC341229	100%	97%
				*SRK*-18	LC341228	97%	97%
		*SLG* (1201 bp)	BLASTn	** *SLG* ** **-19**	**LC341238**	**99%**	**55%**
SJ-7~9	S10 (Lim)		BLASTn	***SRK*10 (Lim)**	**AY052585**	**99.9%**	**100%**
		*SRK*-KD (1051 bp)		*SRK*11 (Lim)	AY534533	98%	100%
				*SRK*16 (Lim)	AY534535	98%	100%
				*SRK*20 (Lim)	AY534537	98%	100%
				*SRK*29 (Lim)	AY534541	99%	100%
				*SRK*-22	LC341231	98%	95%
				*SRK*-23	LC341232	98%	99%
				*SRK*-31	LC341234	99%	96%
				*SRK*7	KX961701	98%	78%
				*SRK*10	KX961704	98%	78%
				*SRK*16	KX961710	99%	78%
				*SRK*17	KX961711	98%	78%
		*SLG* (1202 bp)	BLASTn	***SLG*10 (Lim)**	**AY052576**	**99.9%**	**100%**
SJ-10~11	S16 (Lim)	*SRK*-KD (1071 bp)	BLASTn	***SRK*16 (Lim)**	**AY052579**	**99.9%**	**100%**
				*SRK*-22	LC341231	100%	94%
				*SRK*10 (Lim)	AY052585	98%	100%
				*SRK*11 (Lim)	AY534533	98%	100%
				*SRK*20 (Lim)	AY534537	97%	100%
				*SRK*29 (Lim)	AY534541	98%	100%
				*SRK*-23	LC341232	97%	100%
				*SRK*-31	LC341234	98%	94%
				*SRK*7	KX961701	99%	76%
				*SRK*10	KX961704	98%	76%
				*SRK*16	KX961710	99%	76%
				*SRK*17	KX961711	99%	76%
		*SLG* (1168 bp)	BLASTn	***SLG*-22**	**LC341239**	**100%**	**64%**
				*SLG*-7	AB009684	99.9%	100%
SJ-12~14	S18 (Lim)	*SRK*-KD (1108 bp)	BLASTn	***SRK*18 (Lim)**	**AY534536**	**99.9%**	**100%**
				*SRK*-6	LC341226	99.9%	95%
		*SLG* (1235 bp)	BLASTn	***SLG*18 (Lim)**	**AY527401**	**100%**	**100%**
				*SLG*-6	AB009682	100%	100%
SJ-15~17	S4 (Lim)	*SRK*-KD (Not amplified)	-	-	-	-	-
		*SLG* (1002 bp)	BLASTn	***SLG*4 (Lim)**	**AY052577**	**99%**	**100%**
				*SLG*-26	LC341241	99%	99%
				*SLG*-11	LC341236	97%	87%
SJ-18~20	S5 (Lim)	*SRK*-KD (1053 bp)	BLASTn	***SRK*-5 (Wang)**	-	**99%**	**99%**
				***SRK*-9 (Wang)**	-	**99%**	**99%**
				*SRK*6	KP117077	99%	94%
		*SLG* (1004 bp)	BLASTn	***SRK*-5 (Wang)**	**-**	**99%**	**94%**
				*SLG*5 (Lim)	AY052578	97%	100%
				*SLG*21 (Lim)	AY529650	95%	100%
				*SLG*24 (Lim)	AY529651	96%	100%
				*SRK*1	KX961695	96%	100%
				*SRK*-9	AB114851	96%	98%
				*SLG*-9	LC341235	95%	90%
				Rs chr7	OY743213	95,97%	100%
				Rs chr8	LR778317	95,97%	100%
SJ-21	S21 (Lim)	*SRK*-KD (1015 bp)	BLASTn	***SRK*-9 (Wang)**	**-**	**99%**	99%
				***SRK*-5 (Wang)**	**-**	**99%**	**99%**
				*SRK*6	KP117077	99%	96%
		*SLG* (1055 bp)	BLASTn	***SLG*21 (Lim)**	**AY529650**	**100%**	**84%**
				*SLG*-9	LC341235	100%	77%
				Rs chr7	OY743213	100,94%	100%
				Rs chr8	LR778317	100,94%	100%
				Rs *SLG* S13-like	XM_056990347	100%	100%
				*SLG*24 (Lim)	AY529651	96%	84%
				*SLG*5 (Lim)	AY052578	94%	84%
				*SRK*-1	KX961695	94%	100%
				*SRK*-9	AB114851	94%	100%
SJ-22	S26 (Lim)	*SRK*-KD (Not amplified)	-	-	-	-	-
		*SLG* (802 bp)	BLASTn	***SLG*26 (Lim)**	**AY529652**	**100%**	**100%**
				*SLG*-29	LC341242	**100%**	**100%**

**Table 2 plants-13-00725-t002:** List of 9 S haplotype-specific markers (kinase domain of *SRK*, *SLG*) developed in this study.

S haplotype	Primer Set	Forward	Reverse	Expected Size
S1 (Lim)	*SRK*1	KD1-F	KD1-R	665 bp
	*SLG*1	*SLG*1-F	*SLG*1-R	388 bp
S4 (Lim)	*SLG*4	*SLG*4-F	*SLG*4-R	529 bp
S5 (Lim)	*SLG*5	*SLG*5-F	*SLG*5-R	409 bp
S8 (Lim)	*SRK*8	KD8-F	KD8-R	271 bp
	*SLG*8	*SLG*8-F	*SLG*8-R	314 bp
S10 (Lim)	*SRK*10	KD10-F	KD10-R	411 bp
	*SLG*10	*SLG*10-F	*SLG*10-R	369 bp
S16 (Lim)	*SLG*16	*SLG*16-F	*SLG*16-R	216 bp
S18 (Lim)	*SRK*18	KD18-F	KD18-R	105 bp
	*SLG*18	*SLG*18-F	*SLG*18-R	353 bp
S21 (Lim)	*SLG*21	*SLG*21-F	*SLG*21-R	529 bp
S26 (Lim)	*SLG*26	*SLG*26-F	*SLG*26-R	131 bp

**Table 3 plants-13-00725-t003:** Materials of radish breeding lines used in this study.

Accession Number	Type of Fleshy Root	Color of Fleshy Root	Source
SJ-1	Narrow elliptic	Green and White	South Chinese
SJ-2	Acicular	Green and White	South Chinese
SJ-3	Narrow elliptic	Green and White	South Chinese
SJ-4	Oblong	Green and White	South Chinese
SJ-5	Medium elliptic	Green and White	North Chinese
SJ-6	Medium elliptic	Green and White	North Chinese
SJ-7	Bell shaped	Green and White	North Chinese
SJ-8	Oblong	Green and White	South Chinese
SJ-9	Oblong	Green and White	South Chinese
SJ-10	Oblong	Green and White	South Chinese
SJ-11	Oblong	Green and White	South Chinese
SJ-12	Ovate	Green and White	South Chinese
SJ-13	Medium elliptic	Green and White	North Chinese
SJ-14	Medium elliptic	Green and White	North Chinese
SJ-15	Bell shaped	Green and White	North Chinese
SJ-16	Oblong	White	South Chinese
SJ-17	Oblong	White	South Chinese
SJ-18	Bell shaped	Green and White	North Chinese
SJ-19	Oblong	Green and White	South Chinese
SJ-20	Oblong	Green and White	South Chinese
SJ-21	Bell shaped	Green and White	South Chinese
SJ-22	Narrow elliptic	Green and White	North Chinese

**Table 4 plants-13-00725-t004:** PCR condition for class I, II universal primer sets used in this study.

Gene	Forward	Reverse	Annealing Temperature/Cycle
Class I *SRK*-KD	UV*SRK*-F	UV*SRK*-R	54 °C/34X
Class I *SLG*	*SLG*-I-F	*SLG*-I-R	50 °C/39X
	PS22	*SLG*-I-R	54 °C/35X
Class II *SRK*-KD	KS2	KA2	61 °C/35X
Class II *SLG*	*SLG*-II-F	*SLG*-II-R	55.5 °C/39X
	Rs9*SLG*-F	Rs9*SLG*-R	53 °C/35X
	UV*SLG*II-F	UV*SLG*II-R	61 °C/30X

## Data Availability

Upon request, the corresponding author can provide access to the research data generated in this study.

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
