# Peer review of "Development of S Haplotype-Specific Markers to Identify Genotypes of Self-Incompatibility in Radish (*Raphanus sativus* L.)"

_plants, 2024, doi:10.3390/plants13050725_

Round 1
Reviewer 1 Report
Comments and Suggestions for Authors
The authors used previous sequences to clone and compare sequence with differnt self-incompatibility domains. Furthermore, they classified different types of self-incompatibility and developed some molecular markers to distinguish F1 purity. Overall, the experimental design is simple and easy to understand and the originality is not high because the methods and data are not new. I have some suggestions listed below. First, the logic of INTRODUCTION is poor. For example, the first paragraph described general information of radish. The second paragraph switched to the general information of self-incompatibility. But how to linke between them? Furthermore, the background did not describe the previous investigations fully. I advise the authors to enrich this section. The Disscussion section is weak, please deepen your results as compared to previous investigations but not only the present study.
Author Response
Dear Reviewer No. 1,
Thank you for your valuable comments and suggestions. We carefully considered your feedback and implemented several revisions in the paper.
The revised manuscript, along with feedback, has been uploaded, and the modified content is highlighted in red text.
For detailed explanations regarding the modifications, please see the attachment.
We express our gratitude for your invaluable feedback.
Best regards,
Seong Ho Heo

Reviewer 2 Report
Comments and Suggestions for Authors
Self-incompatibility (SI) is a mechanism in plants that prevents self-cross and promotes outcrossing to maintain genetic diversity. The SI system has been widely applied to F1 hybrid breeding, including radish. By utilizing the SI system in radish, it would be beneficial in reducing breeding time and aiding in the development of novel F1 varieties. Previous PCR methods like PCR-RFLP and SCAR still have some difficulties in identifying S haplotypes in radish. In the paper “Development of S haplotype-specific markers to identify genotypes of self-incompatibility in radish (Raphanus sativus L.)”, Heo et al. developed radish S haplotype-specific markers based on the nucleotide sequences of SRK and SLG using 22 elite radish inbred lines. These S haplotype-specific markers developed in this study have the potential to be used for the radish F1 hybrid breeding process. This is a very interesting paper, and it fits the journal well. The experiments were well-designed and executed, and the conclusions were appropriate for the presented results. I only have one minor question: since radish has self-incompatibility, how were the 22 inbred radish lines made? In other words, how did the author overcome the SI to perform the self-cross? Forgive me if the question is too simple, but I checked the entire paper thoroughly and couldn't find any clues.
Author Response
Dear Reviewer No. 2,
It is more encouraging after seeing your detailed commands and thank very much for your precious comments.
Comments 1: how were the 22 inbred radish lines made? In other words, how did the author overcome the SI to perform the self-cross?
Response 1: The 22 inbred radish lines were made through self-pollination more than 7 generations to stably fix commercially valuable genetic traits (root color, shape, etc.) and these radish lines has been used as parental lines for radish F1 hybrid breeding in East Asia (Korea, China, Japan). To breed these 22 inbred radish lines and made genetic traits homozygous, we overcame self-incompatibility through two methods and performed self-crossing.
Firstly, during self-pollination each inbred radish lines, high-concentration carbon dioxide (CO2) gas was treated to disrupt the SRK-SP11 recognition reaction occurring on the stigma surface. Inhibiting the SRK-SP11 recognition reaction prevents the activation of protein kinase (SRK), avoiding signal cascade triggered by its activation. This disruption ensures normal growth of the self-pollen tube as essential factors for self-pollen germination are not degraded, leading to successful self-crossing and seed setting.
Secondly, we performed bud pollination for each generation. The most active period for production of the substance that inhibits the growth of a self-pollen tube (self-incompatibility substances) is during the flowering stage; no self-incompatibility substances are generated during bud or senescent flower stages. Thus, before the flowering stage, we manually opened the buds using forceps, applied anther onto the stigma, and performed self-pollination. Since self-incompatibility did not occur, the self-pollen tube developed normally, allowing for successful inbreeding and self-crossing.
In summary, self-incompatibility was overcome during the breeding of the 22 inbred radish lines through high-concentration carbon dioxide (CO2) gas treatment and bud pollination. These methods were repeated for each generation until genetic traits were stably homozygous (at least up to the 7th generation).
Please inform me if any revisions require further clarification or explanation. Thank you.
Best regards,
Seong Ho Heo
Round 2
Reviewer 1 Report
Comments and Suggestions for Authors
I appreciate the authors' efforts on the improvement of manuscript. All questions as I advised has been addressed by authors. I agree to publish this manuscript.